# Searching Efficient Semantic Segmentation Architectures via Dynamic Path Selection

**Yuxi Liu**[1,2*]   **Min Liu**[1,2*]   **Shuai Jiang**[1,2]   **Yi Tang**[3†]   **Yaonan Wang**[1,2]

[1]School of Artificial Intelligence and Robotics, Hunan University
[2]National Engineering Research Center of Robot Visual Perception and Control Technology
[3]Department of Data and Systems Engineering, The University of Hong Kong
yuxi_liu@hnu.edu.cn   yiitang@hku.hk

## Abstract

Existing NAS methods for semantic segmentation typically apply uniform optimization to all candidate networks (paths) within a one-shot supernet. However, the concurrent existence of both promising and suboptimal paths often results in inefficient weight updates and gradient conflicts. This issue is particularly severe in semantic segmentation due to its complex multi-branch architectures and large search space, which further degrade the supernet's ability to accurately evaluate individual paths and identify high-quality candidates. To address this issue, we propose Dynamic Path Selection (DPS), a selective training strategy that leverages multiple performance proxies to guide path optimization. DPS follows a stage-wise paradigm, where each phase emphasizes a different objective: early stages prioritize convergence, the middle stage focuses on expressiveness, and the final stage emphasizes a balanced combination of expressiveness and generalization. At each stage, paths are selected based on these criteria, concentrating optimization efforts on promising paths, thus facilitating targeted and efficient model updates. Additionally, DPS integrates a dynamic stage scheduler and a diversity-driven exploration strategy, which jointly enable adaptive stage transitions and maintain structural diversity among selected paths. Extensive experiments demonstrate that, under the same search space, DPS can discover efficient models with strong generalization and superior performance.

## 1   Introduction

Semantic segmentation is a fundamental task in computer vision which focuses on pixel-level classification of images. In recent years, deep neural networks have achieved impressive success in semantic segmentation [1, 2, 3]. A critical factor behind this progress is the design of network architectures, which greatly impact overall performance. However, designing architectures that are both accurate and efficient often require substantial expertise and iterative experimentation.

To address this challenge, neural architecture search (NAS) has been proposed to automatically discover optimal architectures and reduce the need for manual effort. Current NAS methods [4, 5, 6, 7, 8, 9] typically embed the search space into a one-shot supernet and optimize it through weight sharing. Compared to traditional NAS methods, which usually require training thousands of networks, one-shot NAS significantly reduces the computational cost while having superior performance.

Nevertheless, the search space in NAS is enormous, especially for multi-branch segmentation networks (e.g., FasterSeg [8], with a search space of $10^{58}$). Such space makes it difficult for the supernet's parameters to adapt to different subnets (paths) [10], leading to gradient oscillations [11].

---

[*]Equal contribution.
[†]Corresponding author.

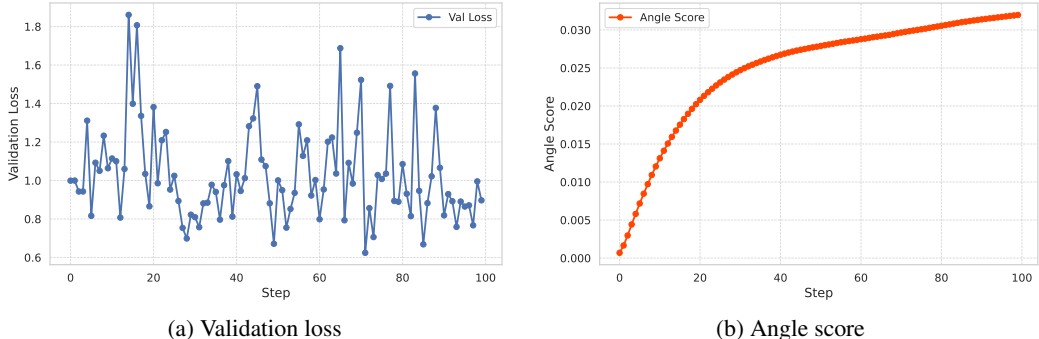

(a) Validation loss          (b) Angle score

Figure 1: Comparison of different performance proxies on a fixed path during the initial stage of supernet training. (a) Validation loss. (b) Angle score.

Additionally, the simultaneous presence of suboptimal and promising paths in the supernet further complicates the optimization process, as weak paths interfere with the highly shared weights [12].

To avoid these issues, some methods [13, 14] introduce a multi-path sampler, which samples multiple paths simultaneously during supernet training and filters out poor-performing ones, prioritizing training on paths with higher quality. These approaches solely rely on the validation loss to determine whether a path is promising. For example, GreedyNAS [13] computes the loss on only 2% of the validation set and use it as an evaluation metric. However, the limited sample size leads to high variance in performance estimates, making this metric unreliable—especially during the early stages of training. As shown in Fig. 1a, we randomly sampled a path during supernet training and observed significant fluctuations in its validation loss in the early stage, a pattern consistently found across other paths. While using a larger portion of the validation set could help reduce this variance, it would also incur substantial computational costs. Furthermore, relying solely on validation loss fails to capture a network's full potential from multiple perspectives. As suggested by recent study [15], the design of effective proxies should account for three key factors: expressiveness, generalization performance, and convergence.

In this paper, we have proposed corresponding proxies to quantify these three factors. For convergence, inspired by *ease-of-convergence hypothesis*[3], we adopt the angle score [17, 18], a metric that measures the distance between initialized and trained weights, to indicate the convergence behavior of a given path. As shown in Fig. 1b, this metric provides a more stable signal than validation loss, exhibiting a steady increase throughout training. For expressiveness estimation, an accuracy predictor is employed to generate relevant scores. Finally, leveraging insights from information bottleneck (IB) theory [19], we design a multi-scale IB (MS-IB) objective function tailored for segmentation models to assess their generalization capacities.

However, all three proxies cannot be uniformly utilized throughout supernet training, as their importance and reliability evolve across different training phases. For instance, the angle score tends stabilizes in the mid-to-late stages, as most paths have already converged by then, making it less discriminative for evaluating path quality. Similarly, the accuracy predictor used for expressiveness estimation must be trained on representative path samples. Nevertheless, in early training stages, the supernet produces unstable and noisy paths, which lead to unreliable training data for the predictor. As a result, accurate expressiveness estimation only becomes feasible once the supernet has reached a more stable phase. (Experimental evidence can be found in Appendix).

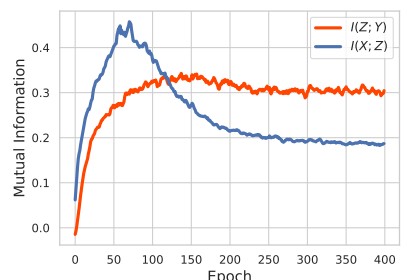

Figure 2: Changes in mutual information during supernet training.

Moreover, according to the IB theory [19], a well-generalized model should compress the input while preserving only the task-relevant features necessary for accurate prediction. To characterize this process, we monitor the mutual information between the input and intermediate representations (i.e., feature maps), denoted as $I(X; Z)$, and between the representations and the output, denoted

---

[3]A strong correlation exists between rapidly converging architectures and high-performing ones [16].

as $I(Z;Y)$. As shown in Fig. 2, $I(X;Z)$ first increases and then declines during early training, suggesting that the model first retains input information to fit the data, and subsequently compresses irrelevant features to improve generalization. At this point, the model has not started compressing redundant information, making MS-IB a relatively less accurate proxy for evaluating generalization.

Building on these empirical observations, we present a novel dynamic path selection (DPS) strategy that progressively shifts focus across these proxies throughout different stages of supernet training. Specifically, our method emphasizes convergence in the first stage, expressiveness in the middle stage, and a balanced combination of expressiveness and generalization in the final stage. As an additional practical consideration, FLOPs is included as a model complexity proxy across all training stages.

To realize this multi-stage, multi-objective framework, DPS integrates three key components. First, path selection is based on Pareto optimality, identifying non-dominated paths that achieve favorable trade-offs among multiple objectives. Second, a dynamic stage scheduler adaptively transitions between stages based on improvement trends and dynamic thresholds, eliminating the need for manual scheduling. Finally, to prevent the path search from collapsing into a narrow search space, we propose a diversity-driven exploration strategy that encourages sampling of structurally diverse paths.

Our main contributions are summarized as follows:

- We propose DPS, a selective training strategy for one-shot NAS in semantic segmentation, which dynamically shifts the selection focus across convergence, expressiveness, and generalization throughout supernet training. This stage-wise design enables comprehensive path evaluation and focuses optimization on high-quality paths in an adaptive manner.

- Three distinct proxies are proposed to evaluate path quality from multiple perspectives, aligning with the three stages of DPS. These metrics provide explicit feedback signals that guide the supernet training, helping to identify and optimize for promising paths during the search process.

- Extensive experiments demonstrate that DPS achieves state-of-the-art results within the same search space. The architectures discovered by DPS exhibit strong generalization, deliver superior performance, and maintain high efficiency.

## 2 Related Work and Background

In one-shot NAS, the architecture search space $\mathcal{A}$ is encoded into an over-parameterized supernet $\mathcal{N}(\mathcal{A}, W)$, where $W$ denotes the set of shared weights across all candidate paths. The supernet is trained using a weight-sharing strategy and the corresponding training objective can be formulated as:

$$W_{\mathcal{A}} = \arg\min_{W} \mathcal{L}_{\text{train}}(\mathcal{N}(\mathcal{A}, W)), \tag{1}$$

where $\mathcal{L}_{\text{train}}$ is the loss function on the training set. Once the supernet has converged, the optimal path $a^*$ can be obtained from the supernet:

$$a^* = \arg\max_{a \in \mathcal{A}} \text{Acc}_{\text{val}}(\mathcal{N}(a, w_a)), \tag{2}$$

where $w_a$ is the weight inherits from $W_{\mathcal{A}}$ and $\text{Acc}_{\text{val}}$ denotes the performance evaluated on the validation set.

As summarized in [20], one-shot NAS approaches mainly fall into two categories, distinguished by the way supernets are represented and optimized. The following provides an overview of each.

**Gradient-based.** Gradient-based NAS methods, such as DARTS [4], perform joint optimization of supernet parameters and weights through gradient descent within a differentiable search space. Due to their efficiency and end-to-end nature, most NAS approaches for semantic segmentation [7, 21, 22, 8, 9, 23] adopt this gradient-based paradigm. However, such joint optimization complicates the training of the supernet and inevitably introduces biases that may mislead the architecture search [6]. Moreover, these methods suffer from high GPU memory consumption [13] and struggle to incorporate different architecture constraints (e.g., FLOPs, latency and energy consumption) during the search process [6]. Instead, they often turn to relaxed regularization terms [24, 5], which offer limited efficacy in ensuring strict compliance with various constraints.

**Sampling-based.** Sampling-based NAS typically search on a discrete search space, and decouple the supernet training and architecture search into two separate stages. In the first stage, supernet was trained using different sampling strategies, such as uniform sampling [6, 25] or multi-path sampling [26, 27, 13, 14, 12] with different priorities. During this process, each sampled path is optimized individually, which alleviates the weight coupling issue observed in gradient-based NAS. In the second stage, typical search algorithms (e.g., evolutionary algorithms) are applied to identify the optimal path. Unlike gradient-based NAS, where hard constraints are difficult to enforce, sampling-based NAS can flexibly incorporate various resource constraints into different search algorithms.

## 3 Method

In this section, we will present our sampling-based NAS method named DPS. First, we begin with introducing different proxies to evaluating convergence, expressiveness, and generalization capacity of a candidate path. Then we discuss the path selection process in detail, which includes three main components: (1) a pareto optimality-based selection strategy, (2) a diversity-driven exploration mechanism, and (3) a dynamic stage scheduling approach for supernet training. Finally, the search pipeline based on evolution algorithm will be explained in short.

### 3.1 Proxies for Path Evaluation

**Angle Score.** To ensure a steady path selection in the early stage of supernet training, inspired by the *ease-of-convergence hypothesis*, we introduce an angle score [17, 18] to evaluate path convergence. Specifically, we compute the angle between the initial weights $\boldsymbol{w_0}$ and the trained weights $\boldsymbol{w}$ for path $a$:

$$\text{angle}(a) = \arccos\left(\frac{\langle \boldsymbol{V}(a, \boldsymbol{w_0}), \boldsymbol{V}(a, \boldsymbol{w})\rangle}{\|\boldsymbol{V}(a, \boldsymbol{w_0})\|_2 \cdot \|\boldsymbol{V}(a, \boldsymbol{w})\|_2}\right) \cdot \frac{1}{(1 + \lambda(t) \cdot n_a)^\alpha}, \quad (3)$$

where $\boldsymbol{V}(a, \boldsymbol{w_0})$ and $\boldsymbol{V}(a, \boldsymbol{w})$ are the concatenated weight vectors, obtained by flattening and stacking all weights along the path from input to output. However, in the early stages of training, certain paths may be sampled more frequently due to the stochastic nature of the search algorithm. This imbalance leads to overestimated angle scores for those paths—not because they converge better, but simply because they are updated more often. Such unfair comparisons can mislead the search process and favor paths with initialization advantages or those sampled more frequently due to randomness.

To mitigate this effect, we incorporate a sampling penalty term that decays with training steps, thereby reducing the impact of early-stage randomness and sampling imbalance on the convergence assessment. Here, $n_a$ denotes the sum of sampling counts for all individual components (e.g., operations, connections) along path $a$, rather than the number of times the entire path is sampled—which is practically negligible due to the enormous search space (e.g., $10^{58}$). The decay factor $\lambda(t)$ is a function of training step $t$, and it follows a linear decay schedule. The parameter $\alpha$ controls the overall penalty intensity.

**Performance Predictor.** To achieve accurate assessment of path expressiveness while avoiding the high computational cost of validation set evaluation, we train a performance predictor once the search process enters the second stage (e.g., 200th epoch). The following are the training details: (1) we begin by randomly sampling 1000 paths and measuring their actual accuracy through the current supernet. (2) we then split these samples into training and test sets at an 8:2 ratio. (3) Finally, we train a random forest regressor [28] on the training set and evaluate its performance on the test set. The regressor is configured with 100 decision trees and no maximum depth limitation.

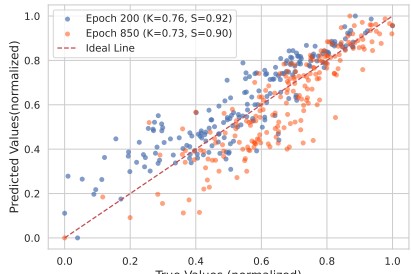

Figure 3: Rank correlation between the predicted values and the true values.

To evaluate the reliability of the performance predictor, we report Kendall's $\tau$ and Spearman's $\rho$ computed on the test set. Furthermore, to ensure that the regressor maintains strong predictive capability in later stages, we randomly sample 200 paths at the 850th training epoch of the supernet

and measure their actual accuracy. The previously trained regressor (from the 200th epoch) is then used to predict their performance. As shown in Fig. 3, the random forest regressor achieves high rank correlation in both early and late stages (e.g., K = 0.76 / S = 0.92 at epoch 200; K = 0.73 / S = 0.90 at epoch 850), indicating consistent and reliable performance throughout the search process.

**Information Bottleneck.** Generalization is a fundamental property of deep neural networks (DNNs), as it determines how well a model performs on unseen data. According to the IB theory [19], DNNs achieve strong generalization by progressively compressing the input information and discarding task-irrelevant redundancy. This compression encourages the learning of compact and informative representations, which are less prone to overfitting and more likely to generalize well. To explicitly model this trade-off between information compression and task relevance preservation, the IB framework introduces a formal objective function:

$$\mathcal{L}_{IB}[p(z|x)] = I(Z;Y) - \beta I(X;Z), \tag{4}$$

where $I(X;Z)$ measures the mutual information between the input $X$ and the learned representation $Z$, and $I(Z;Y)$ captures the informativeness of $Z$ with respect to the target output $Y$. The parameter $\beta$ controls the trade-off between compression and prediction. Mutual information $I(X;Y)$ can be defined as the Kullback-Leibler divergence between the joint distribution $p_{X,Y}$ and the product of marginal distributions $p_X \otimes p_Y$:

$$I(X;Y) = D_{KL}(p_{X,Y}\|p_X \otimes p_Y) = \mathbb{E}_{p_{X,Y}}[\log \frac{p_{X,Y}(x,y)}{p_X(x)p_Y(y)}]. \tag{5}$$

This theoretical foundation motivates us to leverage the IB principle as a measure of generalization in our DPS framework. Specifically, by calculating the mutual information between input images $X$ and feature maps $Z$, as well as between feature maps $Z$ and output label $Y$, we construct an IB-based proxy that can select paths with better generalization capabilities. However, it is impossible to acquire exact distribution of $X$, $Y$ and $Z$.

Inspired by [29, 30], we approximate $\mathcal{L}_{IB}[p(z|x)]$ by estimating a lower bound of $I(X;Z)$ and an upper bound of $I(Z;Y)$. This choice aligns with the IB objective: the lower bound on $I(X;Z)$ avoids overestimating compression and losing informative input features, while the upper bound on $I(Z;Y)$ prevents overestimating task relevance, ensuring sufficient predictive information is retained. As mentioned in [29], KL divergence has a dual representation known as the Donsker-Varadhan representation:

$$D_{KL}(P\|Q) = \sup_{T:\Omega\to\mathbb{R}} \left(\mathbb{E}_P[T] - \log(\mathbb{E}_Q[e^T])\right), \tag{6}$$

where $T$ is any function from the space $\Omega$ to $\mathbb{R}$ such that the expectations are finite. To make the problem tractable, we parametrize the function $T$ using a neural network, denoted as $T_\theta$. Thus, $I(Z;Y)$ can be estimated as:

$$I(Z;Y) \geq \mathbb{E}_{p_{Z,Y}}[T_\theta] - \log(\mathbb{E}_{p_Z \otimes p_Y}[e^{T_\theta}]). \tag{7}$$

The network parameter $\theta$ are optimized by maximizing the objective function using gradient ascent (see Appendix A for details).

Having estimated the dependency between feature maps and labels, we now turn to the second term of the IB objective function. Since the direct computation of mutual information is generally intractable due to the unknown joint distribution $p_{X,Z}$, we adopt a variational approach known as variational Contrastive Log-ratio Upper Bound (vCLUB) [30], which provides a tractable upper bound of the mutual information. Specifically, it introduces a variational distribution $q_\phi(z|x)$ to approximate the true conditional distribution $p(z|x)$, and derives the following inequality:

$$I(X;Z) \leq \mathbb{E}_{p_{X,Z}}[\log q_\phi(z|x)] - \mathbb{E}_{p_X \otimes p_Z}[\log q_\phi(z|x)]. \tag{8}$$

In this formulation, the first expectation is taken over the true joint distribution $p_{X,Z}$, while the second is over the product of marginals $p_X \otimes p_Z$. The variational estimator $q_\phi(z|x)$ is implemented as a neural network parameterized by $\phi$, and is trained to tighten the bound by maximizing the right-hand side of Eq. 8 (see Appendix A for details).

**Multi-Scale Information Bottleneck (MS-IB).** Based on Eq. (7) and Eq. (8), we implement an effective estimation of the Information Bottleneck (IB) objective. Nevertheless, this estimation is performed on a single-scale feature representation, often extracted from the final layer before the segmentation head.

For dense prediction tasks such as semantic segmentation, relying solely on the final representation $Z$ may be suboptimal. This task requires capturing both fine-grained local details and high-level global context. Modern segmentation architectures [31, 32, 2, 33] typically exploit multi-scale feature maps extracted from different stages of the backbone network. Only using the final representation $Z$ will neglects the potential redundancy or relevance encoded in other features with different scales.

To address this limitation, we propose the MS-IB, which extends the IB estimation to multiple stages of the segmentation network:

$$\mathcal{L}_{\text{MS-IB}} = \sum_{s=1}^{S} \left( I(Z_s; Y) - \beta I(X; Z_s) \right), \tag{9}$$

where $S$ denotes the total number of stages considered. This formulation enables a more comprehensive evaluation of the generalization capacity of different paths.

## 3.2 Dynamic Path Selection

**Path Selection with Pareto Optimality.** We formulate path selection as a multi-objective optimization problem and adopt Pareto optimality to identify non-dominated candidates. At every selection step (every $k$ iterations), we randomly sample $m$ candidate paths from the supernet and evaluate them across multiple objectives—including expressiveness, convergence, and generalization ability—using proxies defined in 3.1. Computational cost (FLOPs) is also considered as an additional constraint. We subsequently determine the set of non-dominated paths among the candidates. A path $a_i$ is said to *dominate* another path $a_j$ if and only if:

$$\begin{aligned} \forall x \in c, e, g, f, & \quad x_{a_i} \leq x_{a_j}, \\ \exists x \in c, e, g, f, & \quad x_{a_i} < x_{a_j}, \end{aligned} \tag{10}$$

where $c$, $e$, $g$, and $f$ denote the proxies for convergence, expressiveness, generalization, and negative FLOPs, respectively. The set of non-dominated paths forms the Pareto front, representing architectures that achieve optimal trade-offs among the competing objectives. Note that in Eq. (10), the metric $f$ is used consistently throughout the entire supernet training process, while the other metrics are activated only during their respective stages.

Our objective is to select the top-$k$ paths from the Pareto front. If the number of non-dominated candidates is fewer than $k$, we supplement the selection with additional high-quality paths, chosen based on a composite score calculated by summing their ranks across all proxies.

**Diversity-driven Exploration.** To prevent the path selection from collapsing into narrow and suboptimal regions of the search space, we introduce a diversity-driven exploration strategy that promotes structural diversity and balances exploitation with exploration.

Let $\mathcal{S} = \{a_1, a_2, \ldots, a_m\}$ be the set of $m$ candidate paths sampled at each training iteration, and let $\mathbf{h}_i$ denotes the discrete structural encoding of path $a_i$, we define the *pairwise structure distance* between two paths $a_i$ and $a_j$ using a weighted Hamming distance:

$$\text{Dist}(a_i, a_j) = \sum_{l=1}^{L} \omega_l \cdot \mathbb{I}[\mathbf{h}_i^{(l)} \neq \mathbf{h}_j^{(l)}], \tag{11}$$

where $L$ is the maximum encoding length (with padding applied as needed), $\omega_l$ is a weight for the $l$-th position and $\mathbb{I}[\cdot]$ represent the indicator function. Then we estimate a diversity score of each path by computing its average *pairwise structure distance* within the sampled set:

$$\text{diversity}(a_i) = \frac{1}{m-1} \sum_{j=1}^{K} \text{Dist}(s_i, s_j), \tag{12}$$

Finally, the path with the highest diversity score is considered for replacement. If its score exceeds a predefined threshold $\eta$, it replaces the worst-performing path in the top-k set to maintain diversity while preserving performance.

**Dynamic Stage Scheduling.** To maximize the utility of each proxy and facilitate effective, adaptive path selection, we propose a dynamic stage scheduling mechanism that prioritizes different proxy

metrics at different training stages. Specifically, the mechanism focuses on convergence in the first stage, expressiveness in the middle stage, and both expressiveness and generalization in the final stage.

Unlike methods that enforce rigid transitions between training phases, our approach dynamically adjusts the selection focus based on learning dynamics. Stage transitions are determined by monitoring the improvement trend of the current main metric over a sliding window of size $t_{win}$. A transition is triggered when the average improvement drops below a dynamic threshold:

$$\tau = \epsilon \cdot \mathrm{std}(\mathrm{history}[-t_{win} :]), \tag{13}$$

and remains non-negative. Here, $\mathrm{history}[-t_{win} :]$ denotes the most recent $t_{win}$ values of the current metric, and $\epsilon$ is a sensitivity coefficient that controls the responsiveness of the threshold to historical variation. Note that, in stage II, the performance predictor is used solely for path ranking and does not provide estimates of relative improvement over time. Therefore, we instead use the training loss to assess the improvement trend and determine whether a stage transition should occur.

This approach dynamically adjusts the threshold based on the volatility of the metric: high variance delays path switching to allow for further optimization, while low variance enables timely transitions when performance improvements plateau.

The details of overall dynamic path selection strategy is given in the Appendix B.

### 3.3 Evolution Search with Elite Population.

To better initialize the population in the evolutionary algorithm and improve search efficiency, we maintain an elite population $\mathcal{M}$ after entering final stage of the training process. Specifically, in each iteration, the top-$k$ paths are added to this elite population. If the population exceeds the predefined size limit $|\mathcal{M}|$, the oldest individuals are removed to maintain a fixed capacity. This strategy ensures that high-quality paths are preserved and reused during the search process, thereby accelerating the discovery of optimal paths. The algorithm details of evolution search refer to Appendix C.

## 4 Experiments

### 4.1 Experimental Settings

**Datasets.** We conduct experiments on three widely used semantic segmentation datasets: Cityscapes[34], CamVid [35], and BDD100K[36]. Cityscapes contains 2,975 training, 500 validation, and 1,525 test images with a resolution of $1024 \times 2048$, annotated with 19 semantic classes. CamVid consists of 367 training, 101 validation, and 233 test images at $720 \times 960$ resolution, labeled with 11 categories. BDD100K provides 7,000 training and 1,000 validation images (resolution: $720 \times 1280$), also annotated with 19 semantic classes. All models are trained without external data such as ImageNet [37].

**Search Space.** For a fair comparison, we adopt the same search space from SqueezeNAS [21] and FasterSeg [8]. The first search space use inverted bottleneck [38] as the basic building block and have five different factors, including kernel size, expand ratio, dilation ratio, groups and network depth. The second search space supports multi-branch architecture design. It is based on an $L$-layer cell framework, where each branch operates at a distinct resolution scale. Within each branch, every cell has five different operation choices. The searchable downsample rates are set to $s \in \{8, 16, 32\}$, allowing flexible control of input resolution and feature hierarchy.

**Implementation Details.** The pipeline of our method is divided into three stages: (1) Supernet training with dynamic path selection. (2) Evolution search with elite population. (3) Network retraining. In concrete, once the supernet is fully converged, we will perform an evolution search on it to obtain the optimal path. For more details on the training and search settings, please refer to Appendix C and D.

### 4.2 Ablation Studies

**Searching on same search space.** To demonstrate the effectiveness of DPS, we first compared it with other one-shot NAS method using the same search space, including samlping-based, gradient-based

Table 1: Comparison of one-shot NAS methods on Cityscapes validation set and CamVid test set, categorized into sampling-based (S), gradient-based (G), and training-free (F) approaches; GFLOPs is measured using an input size of 1024 × 2048.

| Method | Type | GFlops | mIoU (%) | |
| --- | --- | --- | --- | --- |
| | | | Cityscapes | CamVid |
| SqueezeNAS Search Space | | | | |
| Random Search | F | 17.32 | 69.8 | 70.7 |
| Uniform Sampling [6] | S | 17.10 | 71.6 | 71.8 |
| Loss Based [13] | S | 12.57 | 72.7 | 73.1 |
| SqueezeNAS [21] | G | 10.86 | 72.4 | 73.2 |
| Ours | S | 10.96 | **74.5** | **73.8** |
| FasterSeg Search Space | | | | |
| Random Search | F | 31.48 | 69.7 | 67.9 |
| Uniform Sampling [6] | S | 31.77 | 71.2 | 69.9 |
| Loss Based [13] | S | 30.64 | 70.9 | 71.8 |
| FasterSeg [8] | G | 28.20 | 73.1 | 71.1 |
| SasWOT [39] | F | 29.34 | 71.3 | 64.3 |
| Ours | S | 29.59 | **73.2** | **72.1** |

and training-free methods. Note that 'Loss Based' refers to using the validation loss as an evaluation metric to filter out suboptimal paths, as done in GreedyNAS [13]. As shown in Table 1, our DPS achieves the best performance on both search spaces. On the SqueezeNAS search space, our method outperforms the original gradient-based SqueezeNAS by 2.1% on Cityscapes and 0.6% on CamVid, while maintaining nearly the same computational cost (FLOPs). For the FasterSeg search space, [8] employed knowledge distillation to further boost accuracy, whereas our approach achieves slightly higher accuracy without relying on extra training strategies. These results demonstrate that DPS achieves an excellent trade-off between accuracy and computational efficiency, thereby showcasing its effectiveness in architecture search.

**Effect of Different Proxies.** To analyze the effect of different proxies, we conduct ablation studies on each component using various proxy combinations. The results are summarized in Table 2. We observe that increasing the number of proxies significantly improves the network performance. Specifically, the combination involving all three proxies achieves the best result, yielding an mIoU of 74.5%. In contrast, relying on a single proxy results in noticeably inferior performance. These findings highlight the importance of evaluating paths from multiple perspectives. By incorporating more proxies into the path selection process,

Table 2: Comparison of different proxy combinations on Cityscapes dataset.

| Proxies | | | Stage | GFlops | mIoU (%) |
| --- | --- | --- | --- | --- | --- |
| $c$ | $e$ | $g$ | | | |
| ✓ | | | 1 | 14.33 | 72.8 |
| | ✓ | | 2 | 13.82 | 72.6 |
| | | ✓ | 3 | 11.91 | 72.8 |
| ✓ | ✓ | | 1→2 | 12.51 | 73.8 |
| ✓ | | ✓ | 1→3 | 12.12 | 73.4 |
| | ✓ | ✓ | 2→3 | 11.28 | 73.1 |
| ✓ | ✓ | ✓ | 1→2→3 | 10.96 | **74.5** |

our method is able to capture complementary signals during training, enabling a more comprehensive assessment of path quality. This, in turn, enhances the effectiveness of the architecture search and leads to better-performing models.

**Effect of Diversity-driven Exploration and Dynamic Stage Scheduler.** We further study the impact of our proposed diversity-driven exploration (DDE) and dynamic stage scheduler (DSS). As summarized in Table 3, both strategies contribute to performance improvement, with DSS showing a more significant effect. This indicates that adaptive stage switching is critical for ensuring the reliability of the evaluation metrics. Using inappropriate fixed-stage switching strategies can greatly reduce metric reliability, potentially leading to suboptimal path selection.

**Effect of MS-IB.** To investigate the impact of MS-IB on model generalization, we design two different experiments. First, we evaluate the model's robustness against various common corruptions and perturbations following the experimental setup in [40]. Specifically, we construct a benchmark containing 16 types of algorithmically generated corruptions across four categories: noise, blur, weather, and digital. Under this setting, we compare the model's performance using single-scale

Table 3: Ablation study of diversity-driven exploration and dynamic stage scheduler.

| DDE | DSS | GFlops | mIoU |
|-----|-----|--------|------|
|     |     | 10.82  | 72.8 |
| ✓   |     | 11.47  | 73.4 |
|     | ✓   | 10.31  | 73.8 |
| ✓   | ✓   | 10.96  | **74.5** |

Table 4: Ablation study of different IB configurations in transfer learning.

| Method | GFlops | mIoU (%) | |
|--------|--------|--------|--------|
|        |        | CamVid | BDD100K |
| None         | 4.12 | 67.7 | 43.5 |
| Single-scale | 3.72 | **70.2** | 44.9 |
| Multi-scale  | 3.61 | 69.7 | **45.2** |

Table 5: Comparison of model performance under various corruptions across IB configurations.

| Method | Blur | | | | Noise | | | | Digital | | | | Weather | | | | Avg |
|--------|------|------|------|------|------|------|------|------|------|------|------|------|------|------|------|------|------|
|        | Motion | Defoc | Glass | Gauss | Gauss | Impul | Shot | Speck | Bright | Contr | Satur | JPEG | Snow | Spatt | Fog | Frost | |
| None         | 63.1 | 61.3 | 58.0 | 65.1 | 3.4 | 4.3 | 4.1 | 18.7 | 57.9 | 49.3 | 49.4 | 34.7 | **15.6** | 51.1 | 32.2 | 15.0 | 36.4 |
| Single-scale | 64.9 | 63.3 | **58.8** | 66.5 | **4.9** | 7.4 | 5.0 | 17.9 | 58.3 | 50.8 | **53.9** | **38.8** | 14.9 | 50.4 | 31.5 | **18.4** | 37.9 |
| Multi-scale  | **65.0** | **63.7** | 58.6 | **66.9** | 4.8 | **9.9** | **6.3** | **25.9** | **61.1** | **53.4** | 50.7 | 37.9 | 14.4 | **52.9** | **35.3** | 15.3 | **38.9** |

Table 6: Comparison of different NAS methods on Cityscapes. * denotes the model is reduced for acc-efficiency trade-offs, implemented by [41].

| Method | Flops | Resolution | mIoU (%) | |
|--------|-------|------------|------|------|
|        |       |            | Val | Test |
| Auto-DeepLab* [7]    | 27.29 | 512 × 1024  | 71.2 | -    |
| HR-NAS-A [41]        | 1.91  | 512 × 1024  | 74.2 | -    |
| CAS [22]             | -     | 768 × 1536  | 71.6 | 70.5 |
| GAS [42]             | -     | 769 × 1537  | -    | 71.8 |
| FasterSeg [8]        | 28.20 | 1024 × 2048 | 73.1 | 71.5 |
| SqueezeNAS-Large [21]| 10.86 | 1024 × 2048 | 72.4 | -    |
| SasWOT [39]          | 29.34 | 1024 × 2048 | 71.3 | 69.8 |
| Ours                 | 10.96 | 1024 × 2048 | **74.5** | **73.5** |

IB, multi-scale IB, and without the IB module. As presented in Table 5, the multi-scale IB module outperforms the baseline in the majority of corruption types, yielding an average mIoU gain of 2.5%. Meanwhile, the single-scale IB also demonstrates moderate improvements in robustness, with particularly strong performance observed on digital corruptions.

In the second experiment, we evaluate cross-dataset generalization by directly transferring Cityscapes-pretrained models to CamVid[35] and BDD100K [36], fine-tuning only the segmentation head while keeping the backbone frozen. As shown in Table 4, both single-scale and multi-scale IB strategies improve generalization over the baseline. The single-scale IB achieves the highest mIoU on CamVid (70.2%), slightly outperforming the multi-scale version (69.7%). This variant computes the IB based solely on the final feature before the segmentation head, whereas the multi-scale IB aggregates IB scores from multiple feature resolutions (e.g., 1/8, 1/16 of input size), aiming to capture a broader information flow. However, the modest performance drop with multi-scale IB on CamVid suggests that not all feature levels contribute equally to generalization. Since CamVid contains low-resolution images with simpler scenes, additional features may introduce redundancy rather than useful signal. These results indicate that future work could explore adaptive weighting of multi-scale features, emphasizing high-level ones near the output while maintaining lower-level regularization, to better balance informativeness and redundancy.

### 4.3 Comparison to state of the art methods.

In this experiment, we compare our method with existing state-of-the-art NAS approaches on the Cityscapes[34] datasets. As shown in Table 6, DPS achieves superior performance. Specifically, it obtains the highest mIoU on the Cityscapes validation and test sets, reaching 74.5% and 73.5%, respectively.

## 5 Conclusion

We propose DPS, a dynamic path selection strategy for one-shot NAS in semantic segmentation. By integrating stage-wise evaluation based on convergence, expressiveness, and generalization, DPS

adaptively focuses supernet training on high-quality paths, enabling targeted optimization. To align with these three stages, we introduce three complementary proxies that evaluate path quality from corresponding perspectives and guide selection through explicit feedback. Extensive experiments validate its effectiveness in identifying efficient models with strong generalization and performance.

## Acknowledgments and Disclosure of Funding

This work was supported in part by the National Key Research and Development Program of China under Grant 2022YFB3303800, in part by the Natural Science Foundation of Hunan Province under Grant 2024JJ3013, in part by the National Natural Science Foundation of China under Grant 62425305, in part by the Science and Technology Innovation Program of Hunan Province under Grant 2023RC1048.

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

# A    Training Details of Mutual Information Estimator

Before entering the final stage of supernet training, we optimize two mutual information estimators to separately compute the two terms in MS-IB, which are used for evaluating path generalization. Once training reaches the final stage, these estimators are fixed and switched to evaluation mode. The detailed training procedures are shown in Algorithm 1 and Algorithm 2.

---

**Algorithm 1** The optimization algorithm of $T_\theta$.

---

**Input:** Neural network $T_\theta$, training set $\mathcal{D}_{train} = \{(x_1, y_1), (x_2, y_2), \ldots, (x_n, y_n)\}$, number of iterations $T$ required to enter Stage III
**Output:** Optimized neural network $T_\theta$
1: **for** $t = 1$ to $T$ **do**
2:     Sample a mini-batch $\mathcal{B}$ from $\mathcal{D}_{train}$:
3:     $\mathcal{B} = \{(x_1, y_1), (x_2, y_2), \ldots, (x_b, y_b)\}$
4:     Compute $Z$ by the sampled path $a$:
5:     $z_1, z_1, \ldots, z_b = a(x_1, x_2, \ldots, x_b)$
6:     Sample $\overline{Z}$ from marginal distribution of $Z$:
7:     $\overline{z_1}, \overline{z_2}, \ldots, \overline{z_b} \sim p_Z$
8:     Compute the value of $I(Z; Y)$ by Eq.7:
9:     $\mathcal{V} = \frac{1}{b} \sum_{i=1}^{b} T_\theta(\overline{z_i}, y_i) - \log(\frac{1}{b} \sum_{i=1}^{b} e^{T_\theta(\overline{z_i}, y_i)})$
10:    Update the parameters $\theta$ using gradient ascent:
11:    $\theta \leftarrow \theta + \frac{\partial \mathcal{V}}{\partial \theta}$
12: **end for**

---

**Algorithm 2** The optimization algorithm of $q_\phi$.

---

**Input:** Variational distribution $q_\phi(z|x)$, training set $\mathcal{D}_{train} = \{(x_1, y_1), (x_2, y_2), \ldots, (x_n, y_n)\}$, number of iterations $T$ required to enter Stage III
**Output:** Optimized neural network $q_\phi$
1: **for** $t = 1$ to $T$ **do**
2:     Sample a mini-batch $\mathcal{B}$ from $\mathcal{D}_{train}$:
3:     $\mathcal{B} = \{(x_1, y_1), (x_2, y_2), \ldots, (x_b, y_b)\}$
4:     Compute $Z$ by the sampled path $a$:
5:     $z_1, z_2, \ldots, z_b = a(x_1, x_2, \ldots, x_b)$
6:     Sample $\overline{Z}$ from marginal distribution of $Z$:
7:     $\overline{z_1}, \overline{z_2}, \ldots, \overline{z_b} \sim p_Z$
8:     Compute the positive term:
9:     $L_{\text{pos}} = \frac{1}{b} \sum_{i=1}^{b} \log q_\phi(z_i|x_i)$
10:    Compute the negative term:
11:    $L_{\text{neg}} = \frac{1}{b^2} \sum_{i=1}^{b} \sum_{j=1}^{b} \log q_\phi(\overline{z_j}|x_i)$
12:    Compute the vCLUB estimate:
13:    $I_{\text{vCLUB}} = L_{\text{pos}} - L_{\text{neg}}$
14:    Update the parameters $\phi$ using gradient ascent:
15:    $\phi \leftarrow \phi + \frac{\partial I_{\text{vCLUB}}}{\partial \phi}$
16: **end for**

---

# B  Dynamic Path Selection

Details of dynamic path selection are presented below. We summarize the key settings as follows: $t_{win} = 20, m = 10, k = 5, \epsilon_1 = 2e - 5, \epsilon_2 = 4e - 5$.

---

**Algorithm 3** Dynamic Path Selection (DPS)

---

**Input:** Supernet $\mathcal{N}$ with weights $W$, training set $\mathcal{D}_{train}$, performance predictor $f_{pred}$, max iteration $T$, window size $t_{win}$, diversity threshold $\eta$, sensitivity coefficients $\epsilon_1, \epsilon_2$

1: Initialize stage counters: $s = 1$
2: Track historical performance: history $\leftarrow []$
3: **for** $t = 1$ to $T$ **do**
4:     **if** $t \mod 5 == 1$ **then**
5:        Initialize score list: Scores $\leftarrow []$
6:        Sample $m$ paths from search space: $\mathcal{P} = \{a_1, ..., a_{10}\}$
7:        **for** each path $a_i \in \mathcal{P}$ **do**
8:           Compute FLOPs score: $\text{Score}_f(a_i) = -\text{FLOPs}(a_i)$
9:           **if** $s == 1$ **then**
10:             Compute convergence score: $\text{Score}_c(a_i) = \text{AngleScore}(a_i)$
11:           **else if** $s == 2$ **then**
12:             Compute expressiveness score: $\text{Score}_e = f_{pred}(a_i)$
13:           **else**
14:             Compute expressiveness score: $\text{Score}_e = f_{pred}(a_i)$
15:             Compute generalization score: $\text{Score}_g = \text{MS-IB}(a_i)$
16:           **end if**
17:           Scores $\leftarrow$ Scores $\cup \{(a_i, \text{Score}_f, \text{Score}_{c/e/g})\}$
18:        **end for**
19:        $\mathcal{P}_{top_k} \leftarrow \text{NonDominatedSort}(\text{Scores}, k)$
20:        **if** $|\mathcal{P}_{top_k}| < k$ **then**
21:           Supplement $\mathcal{P}_{top_k}$ with paths having lowest aggregated rank across all metrics
22:        **end if**
23:        Apply diversity-driven exploration:
24:        **for** each $a_i \in \mathcal{P} \setminus \mathcal{P}_{top_k}$ **do**
25:           **if** diversity$(a_i) > \delta$ **then**
26:             Identify $a_{\text{worst}} = \arg\max_{a \in \mathcal{P}_{top_k}} \text{RankSum}(a)$
27:             Replace $a_{\text{worst}}$ in $\mathcal{P}_{top_k}$ with $a_i$
28:             **break**
29:           **end if**
30:        **end for**
31:        Train selected paths $\mathcal{P}_{top_k}$
32:        Update history:
33:        **if** $s == 1$ **then**
34:           history $\leftarrow$ history $\cup \left\{ \frac{1}{|\mathcal{P}|} \sum_{a_i \in \mathcal{P}} \text{AngleScore}(a_i) \right\}$
35:        **else if** $s == 2$ **then**
36:           history $\leftarrow$ history $\cup \left\{ \frac{1}{|\mathcal{P}_{top_k}|} \sum_{a_i \in \mathcal{P}_{top_k}} \text{TrainLoss}(a_i) \right\}$
37:        **end if**
38:        Update stage $s$ based on learning dynamics:
39:        Extract recent window: $\mathcal{W} \leftarrow \text{history}[-t_{win} :]$
40:        Compute average improvement: $\Delta = \frac{1}{t_{win}-1} \sum_{i=1}^{t_{win}-1} (\mathcal{W}[i] - \mathcal{W}[i-1])$
41:        **if** $\Delta < \tau$ **then**
42:           $s \leftarrow s + 1$
43:        **end if**
44:     **end if**
45: **end for**

---

## C Evolution Search with Elite Population

Details of the evolutionary search are presented below. We summarize the key settings as follows: $\mathcal{M} = 1,500$, $G = 20$, $N = 50$, $p_m = 0.2$, $p_s = 0.1$, $k = 10$.

---
**Algorithm 4** Evolution Search

---
**Input:** Trained supernet $\mathcal{S}$, search space $\mathcal{A}$, elite population $\mathcal{M}$, number of generations $G$, population size $N$, mutation probability $p_m$, crossover probability $p_c$, validation set $\mathcal{D}_{val}$
**Output:** Optimal Architecture $\alpha^*$
    Initialize population: $\mathcal{P} \leftarrow \text{RandomSample}(\mathcal{M}, N)$
    **for** $t = 1$ to $G$ **do**
        $\mathcal{P}_{parent} \leftarrow$ Select top $k$ subnets from $\mathcal{P}$ by mIoU
        $\mathcal{P}_{child} \leftarrow \emptyset$
        **if** $\text{Random}() < p_c$ **then**
            $\mathcal{P}_{child} \leftarrow \mathcal{P}_{child} \cup \text{Crossover}(\mathcal{P}_{parent})$
        **end if**
        **if** $\text{Random}() < p_m$ **then**
            $\mathcal{P}_{child} \leftarrow \mathcal{P}_{child} \cup \text{Mutation}(\mathcal{P}_{parent})$
        **end if**
        **if** $|\mathcal{P}_{child}| < N$ **then**
            Sample $N - |\mathcal{P}_{child}|$ subnets from $\mathcal{M}$ and add to $\mathcal{P}_{child}$
        **end if**
        $\mathcal{P} \leftarrow \mathcal{P}_{child}$
    **end for**
    $\alpha^* \leftarrow \arg\max_{\alpha \in \mathcal{P}} \text{mIoU}(\mathcal{S}(\alpha), \mathcal{D}_{val})$
    **return** $\alpha^*$

---

## D Implementation Details

**Supernet Training.** In this stage, we train the supernet on Cityscapes[34] dataset. We employ the stochastic gradient descent (SGD) optimizer with an initial learning rate of 0.01, momentum of 0.9, and weight decay of 0.0005. The learning rate is adjusted using a polynomial decay policy with a power of 0.9. During training, we apply standard data augmentation techniques including random cropping, random scaling, and horizontal flipping. All models are trained for 850 epochs with a total batch size of 12 across two RTX 3090 GPUs. In addition, the OHEM loss is adopted to enhance model performance.

**Network Retraining.** For network retraining, we adopt the same training strategy used for the supernet. In the case of transfer learning, the learning rate is set to 0.007, and the models are trained for 500 epochs on CamVid [35] and 200 epochs on BDD100K [36], with all other settings remaining unchanged.

## E Experiments on Proxy Analysis

To examine the behavior of the convergence proxy, we tracked the angle score averaged over 10 randomly sampled paths at different training epochs. As reported in Table 7, the score increases rapidly in the early stage (0.65 at epoch 100) and gradually saturates after epoch 400 (around 1.23 at epoch 800). This confirms that the angle score becomes less discriminative in the mid-to-late stages.

To assess the reliability of the performance predictor across training, we measured its correlation with the ground-truth accuracy at different epochs. Table 8 reports Kendall's $\tau$ and Spearman's $\rho$ correlations. The results indicate that the predictor is relatively noisy in the early stage (e.g., $\tau$=0.53 at epoch 50), but its accuracy improves steadily and stabilizes after around 200 epochs ($\tau$=0.76, $\rho$=0.92). This trend supports our design choice of activating the predictor only from Stage II onward, when the supernet has reached a more stable phase.

Table 7: Angle score across epochs.

| Epoch | Angle Score (avg. 10 paths) |
|---|---|
| 100 | 0.65 |
| 200 | 0.99 |
| 400 | 1.20 |
| 600 | 1.22 |
| 800 | 1.23 |

Table 8: Rank correlations between predicted and true accuracies across epochs.

| Epoch | $\tau$ (Kendall) | $\rho$ (Spearman) |
|---|---|---|
| 50 | 0.53 | 0.72 |
| 100 | 0.63 | 0.82 |
| 200 | 0.76 | 0.92 |
| 400 | 0.75 | 0.92 |

## F  Searched Architectures

Table 9: Searched architecture using DPS on the SqueezeNAS search space. k, g, e, d denote kernel size, groups, expansion ratio, and dilation ratio, respectively.

| Block | Operator | $C_{out}$ | Down |
|---|---|---|---|
| 1 | k3_g2_e1_d1 | 16 | 2 |
| 2 | k5_g1_e3_d1 | 24 | 4 |
| 3 | k5_g2_e1_d1 | 24 | 4 |
| 4 | k5_g1_e3_d1 | 24 | 4 |
| 5 | k3_g1_e1_d1 | 24 | 4 |
| 6 | k5_g1_e3_d1 | 32 | 8 |
| 7 | k3_g1_e3_d2 | 32 | 8 |
| 8 | k3_g1_e1_d1 | 32 | 8 |
| 9 | k5_g1_e3_d1 | 32 | 8 |
| 10 | k5_g1_e6_d1 | 64 | 16 |
| 11 | k5_g1_e6_d1 | 64 | 16 |
| 12 | k5_g1_e3_d1 | 64 | 16 |
| 13 | Identity | 64 | 16 |
| 14 | k3_g1_e6_d1 | 96 | 16 |
| 15 | k3_g2_e1_d1 | 96 | 16 |
| 16 | k3_g1_e3_d1 | 96 | 16 |
| 17 | k3_g1_e3_d2 | 96 | 16 |
| 18 | k3_g1_e1_d1 | 160 | 16 |
| 19 | k3_g1_e1_d2 | 160 | 16 |
| 20 | k5_g1_e3_d1 | 160 | 16 |
| 21 | k3_g1_e3_d2 | 160 | 16 |
| 22 | k3_g1_e1_d2 | 160 | 16 |

Table 10: Searched architecture using DPS on the FasterSeg search space. Left: cells for branch with final downsample rate of 16. Right: cells for branch with final downsample rate of 32.

| Cell | Operator | Exp. | $C_{out}$ | Down |
|---|---|---|---|---|
| 1 | zoomed conv. | 12 | 96 | 8 |
| 2 | zoomed conv. | 6 | 48 | 8 |
| 3 | zoomed conv. | 10 | 80 | 8 |
| 4 | conv. | 4 | 32 | 8 |
| 5 | zoomed conv. | 4 | 64 | 16 |
| 6 | zoomed conv. | 4 | 64 | 16 |
| 7 | zoomed conv. | 8 | 128 | 16 |

| Cell | Operator | Exp. | $C_{out}$ | Down |
|---|---|---|---|---|
| 1 | zoomed conv. | 12 | 96 | 8 |
| 2 | conv. | 6 | 96 | 16 |
| 3 | zoomed conv. ×2 | 4 | 128 | 32 |
| 4 | zoomed conv. ×2 | 4 | 128 | 32 |
| 5 | zoomed conv. | 12 | 384 | 32 |
| 6 | zoomed conv. | 4 | 128 | 32 |

