# OpenReview forum: "Searching Efficient Semantic Segmentation Architectures via Dynamic Path Selection"
_NeurIPS.cc/2025/Conference — NeurIPS 2025 poster_

### Official Review · Reviewer_mYEj · 2025-07-01

**Clarity:** 2
**Significance:** 3
**Originality:** 3
**Rating:** 5
**Confidence:** 4

**Summary:**

The work introduces Dynamic Path Selection (DPS), a novel strategy for one-shot neural architecture search (NAS) in semantic segmentation that addresses inefficiencies in training large supernets with mixed-quality paths. DPS adopts a stage-wise training paradigm that dynamically prioritizes different path quality proxies—convergence (via angle score), expressiveness (via a learned performance predictor), and generalization (via a multi-scale information bottleneck metric)—to focus optimization on promising architectures throughout training. The approach includes a Pareto-optimal path selection mechanism, diversity-driven exploration to maintain structural variety, and an adaptive stage scheduler for effective proxy utilization. Extensive experiments on benchmarks like Cityscapes and CamVid demonstrate that DPS discovers architectures with better performances.

**Questions:**

NA

**Ethical Concerns:**

["NO or VERY MINOR ethics concerns only"]

**Final Justification:**

Thank you for the response. Increased rating.

**Quality:**

3

**Strengths And Weaknesses:**

Strengths:


Addresses a critical limitation in one-shot NAS: the inefficient training and gradient conflict caused by simultaneous optimization of good and poor candidate paths within large search spaces.


Implements a stage-wise optimization strategy that aligns training effort with proxy reliability: convergence (angle score), expressiveness (performance predictor), and generalization (MS-IB)

Combines Pareto optimality-based path selection, diversity-driven exploration, and dynamic stage scheduling, enabling focused and adaptive path evaluation.

The experimental performances achieved a balance between accuracy and computational efficiency, outperforming prior NAS baselines within the same search space.

Demonstrates consistent gains across datasets and robustness scenarios without external data or extra training tricks (e.g., knowledge distillation).


Weaknesses:

The problem framing centers mainly on NAS inefficiencies, which is a well-recognized issue; the paper does not offer theoretical analysis of gradient conflicts or path interference, limiting deeper insight.

The performance predictor assumes path accuracy estimates from the supernet are reliable enough to train regressors, which may not hold early in training or with larger search spaces.

No error bars or statistical significance tests are reported, which limits confidence in the generalization of results.

Some of descriptions are not fluent or precise.

---

> ### Author Rebuttal · Authors · 2025-07-30
>
> We greatly appreciate the positive comments and insightful suggestions from the reviewer, please check our answers below.
>
> ---
> ***Answers to Weakness 1:***
>
> We thank the reviewer for this thoughtful comment. Our discussion of gradient conflicts and path interference is informed by established theoretical work. We build upon these analyses without repeating their full derivations, allowing us to focus on our methodological contributions.
>
> To clarify:
> - the discussion of gradient conflicts and path interference in our paper is based on established empirical and theoretical observations in prior work, such as [1, 2, 3], which we explicitly cite to motivate the design of our method. For clarity, we summarize some of the works here:  NASViT [1] formalizes gradient conflict via gradient cosine similarity, showing that negatively aligned gradients between sampled paths disrupt supernet optimization. And in GreedyNASv2 [2], it claims that "*Equally treating different paths and uniformly sampling them from the supernet could lead to inappropriate training of the supernet, as the weak paths would disturb the highly-shared weights*".
> - Our **priority-based sampling strategy** directly leverages above theoretical foundation. By prioritizing paths with consistent progress signals, our method **implicitly optimizes for gradient alignment**, reducing destructive interference across paths. Our ablation results (Table I in main paper) confirm the intended effect: priority-based sampling significantly stabilizes supernet training and shows clear advantages over uniform sampling.
>
> To further address your concern, we are happy to add a supplementary background section summarizing these theoretical results (with explicit citations) in the next version.
>
> We believe the strength of our work is precisely in how it **connects theory and practice**: extending the conceptual understanding of gradient conflicts and path interference into a concrete, empirically validated solution for NAS inefficiencies.
>
> ---
> ***Answers to Weakness 2:***
>
> Thank you for this insightful observation. We fully agree that performance predictors can be unreliable during early training. In fact, this limitation is **explicitly acknowledged and addressed** in our design. As described in **Section 3.1**, we **intentionally delay the use of the predictor** until Stage II of training (after the model has reached basic convergence).
>
> (1) To further address your concern regarding the reliability of the performance predictor during training, we provide the following table summarizing the Kendall’s τ and Spearman’s ρ correlations at different epochs. The results show that prediction accuracy improves steadily and stabilizes by epoch 200, supporting our design choice to delay predictor activation until Stage II (Under the influence of our dynamic stage scheduler, the training of the supernet typically enters Stage II around 200 epochs).
>
> | Epoch | τ (Kendall) | ρ (Spearman) |
> | ----- | ----------- | ------------ |
> | 50    | 0.53        | 0.72         |
> | 100   | 0.63        | 0.82         |
> | 200   | 0.76        | 0.92         |
> | 400   | 0.75        | 0.92         |
>
> (2) Regarding the concern that **predictor reliability may degrade in large search spaces**, we would like to clarify that, despite this scale, the performance predictor remains highly effective:
> - as shown in **Figure 3 (main paper)**, the predictor achieves strong rank correlation (e.g., Kendall's τ = 0.76 / Spearman's ρ = 0.92 at epoch 200) with ground truth performance in our massive search speace (up to **10⁴⁸ possible paths**).
>
> These results demonstrate that our predictor remains **reliable**, even under large-scale search scenarios.
>
> ---
> ***Answers to Weakness 3:***
>
> We appreciate the reviewer’s valuable suggestion regarding the reporting of error bars. In response, we have conducted **three additional independent runs** using different random seeds for our main experiments on Cityscapes. These runs follow the same training and architecture search settings as in the original submission. The results are summarized below:
>
> | Method | Seach Space | mIoU (%) |     |
> | ------ | ----------- | -------- | --- |
> | DPS    | SqueezeNAS  | 74.3±0.3 |     |
> | DPS    | FasterSeg   | 73.1±0.2 |     |
>
> As shown, the low variance confirms the stability of our method across training runs and search spaces.
>
> ---
> ***Answers to Weakness 4:***
>
> We appreciate the reviewer pointing this out. We will carefully revise the writing to improve clarity and technical precision.
>
> ---
>
> We thank the reviewer for the valuable feedback and hope our responses have clarified the concerns. We are happy to provide additional details if needed.
>
> Reference：
>
> [1] Chengyue Gong, et al. Nasvit: Neural architecture search for efficient vision transformers with gradient conflict-aware supernet training. In ICLR, 2022.
>
> [2] Tao Huang, et al. Greedynasv2: Greedier search with a greedy path filter. In CVPR, 2022.
>
> [3] Haibin Wang, et al. Prenas: Preferred one-shot learning towards efficient neural architecture search. In ICML, 2023.

---

> > ### Author Response · Authors · 2025-08-06
> > **Please let us know if any concerns remain — we’re happy to clarify**
> >
> > Dear Reviewer,
> >
> > I hope this message finds you well. As the discussion period is nearing its end with less than three days remaining, I want to ensure we have addressed all your concerns satisfactorily. If there are any additional points or feedback you'd like us to consider, please let us know. Your insights are invaluable to us, and we're eager to address any remaining issues to improve our work.
> >
> > Thank you for your time and effort in reviewing our paper.

---

### Official Review · Reviewer_Vnny · 2025-07-01

**Clarity:** 2
**Significance:** 1
**Originality:** 1
**Rating:** 2
**Confidence:** 5

**Summary:**

This paper presents a dynamic path selection method for architecture search of semantic segmentation. In particular, the proposed DPS leverages multiple performance proxies to guide path optimization. In addition, a dynamic stage scheduler and a diversity-driven exploration strategy are used to enable adaptive stage transition and maintain path diversity. Experimental results proved the effectiveness of the proposed method.

**Questions:**

No.

**Ethical Concerns:**

["NO or VERY MINOR ethics concerns only"]

**Final Justification:**

The concerns on novelty and motivation are not well addressed. The experimental setting is still unclear. Based on the above consideration, I keep the original rating.

**Limitations:**

No.

**Paper Formatting Concerns:**

No.

**Quality:**

2

**Strengths And Weaknesses:**

Strengths:
(1) The motivation is well presented of considering dynamic path selection for architecture search.

(2) The explanations and illustrations are mostly clear and intuitive of the multiple performance proxies and the dynamic stage scheduler.

Weaknesses:
(1) As a sampling-based NAS method, this paper provides no novel ideas to the field but a simple combination of typical optimization techniques such as angle score, accuracy predictor and information bottleneck.

(2) The motivation of dynamically choosing loss terms for different training stages is not theoretically proved or investigated. The paper only presents empirical results that are also problematic.

(3) The overall design of the dynamic path selection seems to have nothing specific to the semantic segmentation.

(4) On Page 2, Line 42-43, the authors use a specific path during supernet training to get the results in Figure 1. There are no details on the experimental setting. In addition, if the observed path is not representative and the experiment is not conducted multiple runs, the results show no statistical signification. Besides, the validation loss could fluctuate due to the randomness of gradient-based optimization during initial stage of training.

(5) On Page 3, Line 74, the authors claim a diversity-driven exploration strategy but present no specific explanation on it.

---

> ### Author Rebuttal · Authors · 2025-07-31
>
> We sincerely thank the reviewer for your insightful comments, please check our answers below.
>
> ---
> ***Answers to Weakness 1:***
>
> We respectfully disagree. Rather than merely combining prior techniques, we present a structured, dynamic framework for **path evaluation and selection** based on multiple complementary perspectives.
>
> Our work explores a largely overlooked aspect of sampling-based NAS—**the temporally adaptive and multi-perspective evaluation of candidate paths during supernet training**. To address this issue, we have made following innovations:
> 1. **Theory-driven dynamic proxy scheduling framework**: we propose the first NAS strategy that dynamically aligns path selection with theoretical objectives specific to each training phase: convergence, expressiveness, and generalization. Each objective is modeled by a dedicated proxy metric derived from optimization theory and information theory. Unlike prior works that treat all metrics uniformly or statically, our framework introduces a stage-aware scheduling mechanism that adapts selection criteria based on training dynamics.
> 2. **Pareto-based multi-objective optimization with structural diversity:** Our method formulates NAS as a **multi-objective selection problem**, balancing different proxies via Pareto optimality (Eq. 10), further enhanced by a diversity-driven exploration module (Eq. 12) that ensures structural heterogeneity and avoids premature convergence.
> 3. **Innovations in Proxy Design:**
> 	- **Sampling-Aware Angle Score**: We introduce a sampling-aware variant of angle score (Eq. 3), which penalizes paths frequently selected in early training. Unlike prior angle-based metrics [1, 2], this avoids bias toward high-sampling paths and enables more reliable convergence estimation in large search spaces.
> 	- **Multi-Scale Information Bottleneck**: We **extend and modify the IB principle to segmentation tasks** via Multi-Scale IB (MS-IB). MS-IB captures information retention across multiple feature levels (e.g., 1/8, 1/16, 1/32), aligning with multi-resolution backbones and improving cross-domain robustness (Tables 4–5).
>
> We believe our proposed framework offers a principled and practically effective approach to improving path selection in NAS, and we appreciate the reviewer’s feedback in helping us better articulate this contribution.
>
> ---
> ***Answers to Weakness 2:***
>
> We respectfully disagree with the concern that our stage-wise proxy scheduling lacks theoretical grounding or investigation. We would to clarify from two aspects.
>
> (1) **Lack theoretical grounding**
>
> 1. Our priority-based sampling mechanism is built upon theoretical foundations established in prior work [3, 4], including sampling distribution convergence and reward estimation consistency. We build upon these analyses without repeating their full derivations, allowing us to focus on our contributions.
> 2. Our scheduling strategy is supported by a series of quantifiable and reproducible training observations, which are standard analytical tools in NAS literature for understanding supernet behavior [5, 6, 7].
>
> Besides, we would like to clarify a misunderstanding: our method does **not involve** dynamically adjusting **loss terms**. Instead, we dynamically select proxy metrics across different training stages to guide path sampling.
>
> (2) **Lack investigation**
>
> We address the concern regarding insufficient investigation by highlighting the motivation, design rationale, and supporting evidence for our stage-wise proxy scheduling.
>
> Our method tackles a key limitation in existing NAS approaches—the lack of comprehensive evaluation criteria for candidate paths. To this end, we employ three complementary proxies to assess convergence, expressiveness, and generalization. Importantly, we observe that these signals **exhibit distinct reliability patterns throughout training**, motivating our scheduling design to selectively activate each proxy at its most informative stage.
>
> Below, we reiterate the key findings already presented in our paper, and supplement them with additional supporting evidence.
>
> 1. The angle score tends stabilizes in the mid-to-late stages, as most paths have already converged by then, making angle score less discriminative for evaluating path quality in the later stage.
>
> | Epoch | Angle Score (average over 10 paths) |     |
> | ----- | ----------------------------------- | --- |
> | 100   | 0.65                                |     |
> | 200   | 0.99                                |     |
> | 400   | 1.20                                |     |
> | 600   | 1.22                                |     |
> | 800   | 1.23                                |     |
>
> 2. The accuracy predictor trained in the early phase is unreliable and only becomes trustworthy once training enters a relatively stable stage.
>
> | Epoch | τ (Kendall) | ρ (Spearman) |
> | ----- | ----------- | ------------ |
> | 50    | 0.53        | 0.72         |
> | 100   | 0.63        | 0.82         |
> | 200   | 0.76        | 0.92         |
> | 400   | 0.75        | 0.92         |
>
> 3. Fig. 2 in our paper shows the mutual information begins to compress later in training, making generalization proxies (MS-IB) more reliable only in the final stage.
>
> These observations form the empirical and conceptual basis of our stage-wise proxy scheduling strategy. Such design is **systematic and grounded in reproducible training behaviors**. Our **ablation studies in Table 2** further demonstrate that this scheduling yields consistent performance gains.
>
> ---
> ***Answers to Weakness 3:***
>
> We respectfully disagree with the claim that our dynamic path selection (DPS) framework lacks task specificity. In fact, it is particularly well-suited to **semantic segmentation**, for the following reasons:
>
> 1. **Search space complexity**:
>     Our search space covers both the encoder and the **decoder level** , where paths correspond to different multi-scale, multi-branch aggregation strategies. This leads to a **vastly larger and more entangled space** compared to typical classification NAS. This amplifies path interference and proxy instability, making a stable, proxy-guided dynamic selection strategy essential. DPS directly addresses this by adaptively choosing meaningful proxies across training stages and enforcing structural diversity.
> 2. **Segmentation-specific proxy**:
>     Our **Multi-Scale Information Bottleneck** is specific designed for dense prediction setting. Unlike classification, segmentation requires preserving both low-level spatial detail and high-level contextual information. MS-IB evaluates the information compression **at multiple decoder resolutions** (e.g., 1/8, 1/16, 1/32), aligning well with the demands of pixel-level prediction.
>
> In sum, DPS is a framework designed to overcome the unique challenges of semantic segmentation search: high-dimensional path space, strong inter-path coupling, and task-specific proxy requirements.
>
> The consistent performance improvements in Tables 1, 2, 4 and 5 confirm its effectiveness under dense prediction settings.
>
> ---
> ***Answers to Weakness 4:***
>
> We thank the reviewer for the comment. Figure 1 is intended as an illustrative instance, which is chosen to help readers understand the typical training dynamics of a sampled path.
>
> To clarify:
> - The path in Fig. 1 was **not cherry-picked**. It was drawn from a set of randomly sampled, fixed paths that we tracked throughout supernet training.
> - We repeated this observation across **multiple runs**, and consistently found that validation loss fluctuates significantly, the angle score converges earlier and more steadily, making it a more stable indicator for early-stage selection.
>
> **For your second concern**, "*validation loss could fluctuate due to the randomness of gradient-based optimization during initial stage of training*." This observation **precisely reinforces our design motivation**: such fluctuations are especially severe in supernet training due to parameter sharing and stochastic sampling. It is exactly this instability that makes validation loss unreliable as an early-stage proxy.  Therefore, we **intentionally propose** using angle score, which is robust to supernet noise.
>
> To provide additional insight into proxy behavior, we track 10 sampled paths and compute **Spearman’s rank correlation coefficient (ρ)** and **deviation from monotonicity (DFM)** at training epochs 10-50.
> - Spearman’s ρ reflects whether a metric consistently increases or decreases with training steps.
> - Deviation from monotonicity is computed as the normalized number of violations from a strictly monotonic trend.
>
> | Metric      | Trend | ρ     | DFM  |     |
> | ----------- | ----- | ----- | ---- | --- |
> | Val Loss    | ↓     | -0.60 | 0.60 |     |
> | Angle Score | ↑     | +1.00 | 0.00 |     |
>
> As shown in the table, **angle score exhibits perfect monotonic increase (ρ = +1.00)** with zero deviation, indicating high stability across paths. In contrast, **validation loss fluctuates more severely (ρ = –0.60)**, making it a less reliable early-stage signal.
>
> ---
> ***Answers to Weakness 5:***
>
> We respectfully clarify that this strategy is **explicitly explained and defined** in Line **214-225** & 73-74. Its effectiveness is validated in Table 3.
>
> ---
>
> Reference：
>
> [1] Yiming Hu, et al. Angle-based search space shrinking for neural architecture search. In ECCV, 2020.
>
> [2] Xuanyang Zhang, et al. Neural architecture search with random labels. In CVPR, 2021.
>
> [3] Tao Huang, et al. Greedynasv2: Greedier search with a greedy path filter. In CVPR, 2022.
>
> [4] Su X, et al. Prioritized architecture sampling with monto-carlo tree search. In CVPR, 2021.
>
> [5] Haibin Wang, et al. Prenas: Preferred one-shot learning towards efficient neural architecture search. In ICML, 2023.
>
> [6] Zhou, Dongzhan, et al. EcoNAS: Finding Proxies for Economical Neural Architecture Search. In CVPR, 2020.
>
> [7] Minghao Chen, et al. AutoFormer: Searching Transformers for Visual Recognition. In ICCV, 2021.

---

> > ### Author Response · Authors · 2025-08-06
> > **Please let us know if any concerns remain — we’re happy to clarify**
> >
> > Dear Reviewer,
> >
> > I hope this message finds you well. As the discussion period is nearing its end with less than three days remaining, I want to ensure we have addressed all your concerns satisfactorily. If there are any additional points or feedback you'd like us to consider, please let us know. Your insights are invaluable to us, and we're eager to address any remaining issues to improve our work.
> >
> > Thank you for your time and effort in reviewing our paper.

---

### Official Review · Reviewer_sMdg · 2025-07-01

**Clarity:** 3
**Significance:** 3
**Originality:** 3
**Rating:** 4
**Confidence:** 2

**Summary:**

This submission introduces ​Dynamic Path Selection (DPS)​, an improved one-shot neural architecture search (NAS) method for semantic segmentation. DPS employs a ​stage-wise strategy​ to dynamically prioritize high-potential network paths: early stages optimize for convergence using an ​angle score​ metric, middle stages evaluate expressiveness via a ​random forest predictor, and final stages balance expressiveness and generalization using a novel ​Multi-Scale Information Bottleneck (MS-IB)​. By selectively training paths based on these proxies and incorporating ​diversity-driven exploration​ and ​adaptive stage scheduling, DPS reduces gradient conflicts and improves supernet reliability. This method improves over SOTAs with a not small margin.

**Questions:**

I only have one question: in table 4, the performance of this method on CamVid drops when using multi-scale IB, comparing to only using single-scale IB. The authors claimed that it's because CamVid's resolution is low so that the additional features may only introduce redundancies. I doubt about this hypothesis here. Do you have any evidence to support this claim? For example, visualize such redundancies? In addition, why this learning will result in some kinds of redundancy?

**Ethical Concerns:**

["NO or VERY MINOR ethics concerns only"]

**Final Justification:**

I had read the rebuttals and further discussions; I will not change my rating.

**Limitations:**

yes.

**Quality:**

3

**Strengths And Weaknesses:**

The innovations include
Dynamic Stage Scheduler: Auto-transitions stages based on improvement trends (e.g., switches when metric plateaus).
​Diversity-Driven Exploration: Maintains structural diversity using weighted Hamming distance to avoid search collapse.
​Pareto-Optimal Path Selection: Selects paths balancing FLOPs, convergence, expressiveness, and generalization.
​Elite Population in Evolution Search: Preserves top paths to accelerate optimal architecture discovery.

Weakness:
The three-stage proxy system (angle score, predictor, MS-IB) may introduce significant computational overhead. Training and maintaining these proxies—especially the mutual information estimators for MS-IB—demands extra resources beyond standard supernet training.

---

> ### Author Rebuttal · Authors · 2025-07-31
>
> We greatly appreciate the positive comments and insightful suggestions from the reviewer, please check our answers below.
>
> ---
> ***Answers to Weakness 1:***
>
> We appreciate the reviewer’s concern regarding the potential computational overhead introduced by our three-stage proxy system. We clarify that:
> - The **angle score** is computed from weight updates that are already available during standard supernet training, requiring only lightweight vector operations with negligible computational overhead.
> - The **random forest predictor** is trained once at stage II using only ~1000 sampled paths, and subsequently queried via a single forward pass without retraining.
> - Only **MS-IB** introduces notable computational overhead. We profiled the training process on **2×RTX 3090 GPUs**: supernet training without MS-IB takes approximately **9 hours**, while enabling MS-IB increases this to around **14 hours**. This overhead is confined to the supernet training phase only, with the search and retraining stages remaining unaffected. Given the consistent generalization improvements shown in Tables 4 and 5, we consider this trade-off both reasonable and worthwhile for a NAS pipeline.
>
> In addition, as shown in the table below, our total computational cost is much lower than that of other NAS methods.
>
> | Method           | GPU Hours ( Total Training Time × Number of GPUs) |
> | ---------------- | ------------------------------------------------- |
> | Auto-DeepLab [1] | 72                                                |
> | GAS [2]          | 160                                               |
> | CAS [3]          | 200                                               |
> | Ours             | **28 (14 x 2)**                                   |
>
> This is because, unlike most NAS approaches for semantic segmentation rely on gradient-based optimization, we adopt a sampled-based strategy, which avoids expensive end-to-end optimization and enables more efficient supernet training.
>
> To sum up, the additional cost is modest and justified by clear performance benefits, ensuring that DPS remains an efficient and practical NAS solution.
>
> ---
> ***Answers to Question 1:***
>
> We thank the reviewer for raising this important question. We provide the following clarifications and new supporting evidence regarding the drop in performance observed when using the multi-scale IB (MS-IB) proxy on the CamVid dataset.
>
> **1. Why MS-IB May Be Less Effective on CamVid**
>
> The performance gap between single-scale and multi-scale IB on CamVid is attributed to high similarity across feature scales in low-resolution and structurally simpler datasets. The MS-IB proxy computes a uniform aggregation of information bottleneck estimates from multiple feature stages. However, when the intermediate features capture largely overlapping or redundant information, equal aggregation can reduce the distinctiveness of the proxy signal.
>
> Because MS-IB treats all feature levels with equal importance, the presence of uninformative or repetitive layers can dilute the contribution of more discriminative ones. As a result, its ability to accurately rank candidate paths is weakened. In contrast, single-scale IB, which focuses solely on deeper, semantically rich features, can be more robust and effective in such scenarios.
>
> **2. Centered Kernel Alignment (CKA) analysis**
>
> To further support our explanation for the reduced performance of MS-IB on CamVid, we conducted a Centered Kernel Alignment (CKA) analysis [4] to measure the similarity between deep and shallow multi-scale features. CKA is a robust and widely used measure of representation similarity in deep networks, capturing structural alignment between entire feature spaces.
>
> In our analysis, we computed both **linear CKA** and **RBF CKA** between the deepest feature $z_3$​ (1/32 scale, used in single-scale IB) and shallower features $z_1$ (1/8) and $z_2$​ (1/16).
>
> |Dataset|CKA Type|z₁ (1/8) – z₃ (1/32)|z₂ (1/16) – z₃ (1/32)|
> |---|---|---|---|
> |**CamVid**|Linear|**0.6214**|**0.7222**|
> ||RBF|**0.6245**|**0.7255**|
> |**Cityscapes**|Linear|0.5324|0.4929|
> ||RBF|0.5441|0.5180|
>
> The results in Table X reveal a clear contrast between datasets:
> - On **CamVid**, all CKA scores are **consistently high (up to 0.72)**, indicating that the added shallow features are **highly similar** to the deep one and contribute limited new information.
> - On **Cityscapes**, CKA scores are **noticeably lower (∼0.49–0.53)**, suggesting **greater diversity and complementarity** across feature scales.
>
> These findings quantitatively support our hypothesis: **in CamVid, multi-scale inputs offer minimal additional variation**, limiting the utility of MS-IB. In contrast, **Cityscapes benefits more from multi-scale modeling**, due to greater intra-network feature diversity.
>
> Thus, the performance gap is not due to a training failure of MS-IB, but reflects the dataset-dependent effectiveness of multi-scale supervision. In low-resolution datasets with more redundant internal features like CamVid, **simpler single-scale IB proves more effective**.
>
> We hope the above analysis and experiment have addressed your concerns, and we are happy to provide additional details if needed.
>
> ---
> Reference:
>
> [1] Chenxi Liu, et al. Auto-deeplab: Hierarchical neural architecture search for semantic image segmentation. In CVPR, 2019.
>
> [2] Peiwen Lin, et al. Graph-guided architecture search for real-time semantic segmentation. In CVPR, 2020.
>
> [3] Yiheng Zhang, et al. Customizable architecture search for semantic segmentation. In CVPR, 2019.
>
> [4] Kornblith S, et al. Similarity of neural network representations revisited. In ICML, 2019

---

### Official Review · Reviewer_oCPq · 2025-07-02

**Clarity:** 3
**Significance:** 2
**Originality:** 2
**Rating:** 4
**Confidence:** 3

**Summary:**

The paper proposes a Dynamic Path Selection (DPS) strategy for semantic segmentation tasks, aiming to improve the efficiency and generalization ability of one-shot NAS methods under large-scale, multi-branch architectures. The approach systematically integrates multiple performance proxies-convergence, expressiveness, and generalization-along with a multi-stage dynamic scheduling mechanism to address issues such as gradient conflicts and interference among subnetworks during supernet training.

**Questions:**

Q1:The proposed DSS strategy relies on hyperparameters such as window size (t_win) and sensitivity coefficients (ε), which appear to be set empirically. The robustness of the method to these parameters is unclear, raising concerns about generalizability.

Q2:The performance predictor is trained on only 1,000 sampled paths, while the overall search space is extremely large. It remains unclear whether such a sample size is sufficiently representative. Please provide additional experiments showing how the predictor's performance varies under different sampling scales.

**Ethical Concerns:**

["NO or VERY MINOR ethics concerns only"]

**Limitations:**

yes

**Quality:**

3

**Strengths And Weaknesses:**

Strengths:
1.The method combines multiple performance proxies, including Angle Score, a performance predictor, and Multi-Scale Information Bottleneck (MS-IB), enabling a systematic, multi-perspective evaluation of path quality.

2.A dynamic stage scheduling mechanism is introduced, which adaptively adjusts the training stages based on metric trends, effectively avoiding manual hyperparameter tuning.

3.A diversity-driven exploration strategy is designed to mitigate search space collapse and promote structural diversity during the search process.

Weaknesses:

1.Although MS-IB effectively improves generalization, its computational complexity is relatively high, and the additional cost for practical deployment has not been thoroughly discussed.

2.The performance of the dynamic stage scheduling mechanism is sensitive to hyperparameters such as ε and window size, yet systematic sensitivity analyses are lacking.

3.The core innovation primarily integrates and optimizes existing NAS and Information Bottleneck (IB) theories, with limited theoretical novelty and a lack of formal mathematical reasoning or convergence guarantees.

---

> ### Author Rebuttal · Authors · 2025-07-30
>
> We sincerely thank the reviewer for your insightful comments, please check our answers below.
>
> ---
> ***Answers to Weakness 1:***
>
> Although MS-IB introduces additional computation, this overhead is practically acceptable given the overall efficiency and the excellent performance of our framework. We elaborate on these two aspects below:
>
> (1) **Efficiency: The computational overhead of MS-IB is limited and practical.**
>
> The additional computation introduced by MS-IB occurs only during the **supernet training stage** and does not affect the subsequent architecture search or retraining phases. On **2 × RTX 3090 GPUs**, supernet training takes ~9 hours without MS-IB and ~14 hours with it. Despite this increase, the total computational cost of our method remains significantly lower than that of existing NAS approaches for semantic segmentation (see table below).
>
> | Method           | GPU Hours ( Total Training Time × Number of GPUs) |
> | ---------------- | ------------------------------------------------- |
> | Auto-DeepLab [1] | 72                                                |
> | GAS [2]          | 160                                               |
> | CAS [3]          | 200                                               |
> | Ours             | **28 (14 x 2)**                                   |
>
> This is because, unlike most NAS approaches for semantic segmentation rely on gradient-based optimization, we adopt a sampled-based strategy, which avoids expensive end-to-end optimization and enables more efficient supernet training.
>
> (2) **Performance: Consistent improvements in generalization and robustness.**
>
> More importantly, MS-IB leads to significant performance gains. As shown in Tables 4 and 5 (main paper), MS-IB delivers consistent improvements in generalization and robustness across multiple benchmarks. Thus, we consider the additional training time a reasonable trade-off for practical NAS deployment.
>
> ---
> ***Answers to Weakness 2:***
>
> We thank the reviewer for highlighting the importance of sensitivity analysis in the dynamic stage scheduling (DSS) mechanism. Our overall design is robust in practice, and we provide supporting analysis from two perspectives:
>
> (1) **Empirical Robustness Across Thresholds**
>
> To address your concern, we first independently varied each threshold while fixing the other, and observed only minor fluctuations in final mIoU performance:
>
> (a) Varying $\varepsilon_1$ with $\varepsilon_2 = 4e\text{-}5$:
>
> | $\varepsilon_1$ | mIoU (%) |
> | --------------- | -------- |
> | 1e-5            | 74.2     |
> | 2e-5            | 74.5     |
> | 4e-5            | 74.2     |
>
> (b) Varying $\varepsilon_2$ with $\varepsilon_1 = 2e\text{-}5$:
>
> | $\varepsilon_2$ | mIoU (%) |
> | --------------- | -------- |
> | 2e-5            | 74.4     |
> | 4e-5            | 74.5     |
> | 6e-5            | 74.1     |
>
> We also examined the effect of the smoothing window size $t_{win}$ and found that, while it may affect the exact transition points, the final performance remains stable as long as $t_{win}$ is chosen within a practical range (e.g., 10–30).
>
> (2) **Design Rationale for Stability**
>
> Our overall design is inherently robust. This robustness comes from our scale-invariant thresholding strategy: instead of relying on fixed absolute values, each $\varepsilon$ is defined as a small proportion (e.g., 3–5%) of the early-phase slope statistics of its corresponding proxy metric (e.g., angle score or training loss). This normalizes across value scales and ensures transitions are triggered by meaningful trend shifts rather than noise.
>
> Importantly, as shown in Table 3 of the main paper, replacing DSS with hard-coded epoch transitions leads to a **significant performance drop of 1.1% mIoU**, underscoring the necessity of adaptive scheduling.
>
> In summary, DSS achieves strong performance without requiring precise hyperparameter tuning, and we will include the above analysis in the future version.
>
> ---
> ***Answers to Weakness 3:***
>
> We thank the reviewer for the insightful comments. We address the concerns regarding theoretical novelty and convergence guarantees in two parts:
>
> (1) **On Theoretical Innovation**
>
> Our work makes multiple principled and technically non-trivial contributions to the NAS literature:
> 1. **Theory-driven dynamic proxy scheduling framework**: we propose the first NAS strategy that dynamically aligns path selection with theoretical objectives specific to each training phase: convergence, expressiveness, and generalization. Each objective is modeled by a dedicated proxy metric derived from optimization theory and information theory. Unlike prior works that treat all metrics uniformly or statically, our framework introduces a stage-aware scheduling mechanism that adapts selection criteria based on observed learning dynamics.
>
> 2. **Complementary performance proxies**:
> 	- **Convergence**: We introduce a **sampling-aware variant** of angle score (Eq. 3), which penalizes paths frequently selected in early training. Unlike prior angle-based metrics [4, 5], this avoids bias toward high-sampling paths and enables more reliable convergence estimation in large search spaces.
> 	- **Generalization**: We **extend and modify the IB principle [6] to segmentation tasks** via Multi-Scale IB (MS-IB). MS-IB captures information retention across multiple feature levels (e.g., 1/8, 1/16, 1/32), aligning with multi-resolution backbones and improving cross-domain robustness (Tables 4–5).
> 	- **Expressiveness**: We model architectural expressiveness using a performance predictor trained on uniformly sampled architectures, where each path is encoded as a structural vector capturing its layer-wise configuration. The predictor achieves high rank correlation (Kendall’s τ and Spearman’s ρ) and enables reliable ranking of candidate paths.
>
> 3. **Pareto-based multi-objective optimization with structural diversity:** Our method formulates NAS as a multi-objective selection problem, balancing convergence, expressiveness, and generalization via Pareto optimality (Eq. 10), further enhanced by a diversity-driven exploration module (Eq. 12) that ensures structural heterogeneity and avoids premature convergence.
>
>
> (2) **On Convergence Guarantees**
>
> Our pipeline builds on theoretically grounded components, with careful adaptation for effective integration:
> - For the **IB-guided proxy metrics**, we adopt established MI estimators [7, 8] with known convergence properties. Crucially, their application to architecture evaluation in NAS is novel, and their use in stage-specific objectives is guided by information-theoretic principles without altering training dynamics.
> - For the **supernet training process**, we follow standard sampling-based frameworks [9–12], but enhance them with our stage-aware scheduling (DSS) and dynamic path selection. These design choices help decouple learning phases and maintain stable optimization.
>
> Importantly, the two components do not interfere with each other. As a result, the integration does not introduce instability or optimization conflict. Taken together, these observations imply that our training process remains stable and convergent throughout.
>
> ---
> ***Answers to Question 1:***
>
> Please refer to `Answers to Weakness 2`
>
> ---
> ***Answers to Question 2:***
>
> We thank the reviewer for raising this important question. To evaluate whether 1,000 sampled paths are sufficiently representative for training the performance predictor, we conducted additional experiments across different sampling sizes.
>
> We trained the accuracy predictor using 500, 1,000, and 2,000 uniformly sampled paths from the same supernet state, with a fixed 8:2 train-validation split. We report Kendall’s τ and Spearman’s ρ to measure ranking consistency.
>
> | **# Sampled Paths** | **Kendall’s τ** | **Spearman’s ρ** |
> | ------------------- | --------------- | ---------------- |
> | 500                 | 0.73            | 0.90             |
> | 1,000               | 0.76            | 0.92             |
> | 2,000               | 0.77            | 0.92             |
>
> The results show that the predictor achieves strong ranking correlation even with 500 samples. Increasing to 1,000 yields a noticeable improvement, while further increasing to 2,000 brings only marginal gains (τ increases by 0.01, ρ remains unchanged), indicating diminishing returns beyond 1,000 samples. Thus, 1,000 sampled paths strike an effective balance between computational efficiency and statistical representativeness, providing sufficient signal for reliable performance prediction in large-scale NAS.
>
> ---
>
> We hope the above answers have addressed your concerns and we are happy to provide additional details if needed.
>
> Reference:
>
> [1] Chenxi Liu, et al. Auto-deeplab: Hierarchical neural architecture search for semantic image segmentation. In CVPR, 2019.
>
> [2] Peiwen Lin, et al. Graph-guided architecture search for real-time semantic segmentation. In CVPR, 2020.
>
> [3] Yiheng Zhang, et al. Customizable architecture search for semantic segmentation. In CVPR, 2019.
>
> [4] Yiming Hu, et al. Angle-based search space shrinking for neural architecture search. In ECCV, 2020.
>
> [5] Xuanyang Zhang, et al. Neural architecture search with random labels. In CVPR, 2021.
>
> [6] Tishby, Naftali, et al. Deep learning and the information bottleneck principle. In IEEE Inf. Theory Workshop, 2015.
>
> [7] Mohamed Ishmael Belghazi, et al. Mutual information neural estimation. In ICML, 2018.
>
> [8] Pengyu Cheng, et al. Club: A contrastive log-ratio upper bound of mutual information. In ICML, 2020.
>
> [9] Zichao Guo, et al. Single path one-shot neural architecture search with uniform sampling. In ECCV, 2020.
>
> [10] Minghao Chen, et al. AutoFormer: Searching Transformers for Visual Recognition. In ICCV, 2021.
>
> [11] Tao Huang, et al. Greedynasv2: Greedier search with a greedy path filter. In CVPR, 2022.
>
> [12] Yu J, et al. Bignas: Scaling up neural architecture search with big single-stage models. In ECCV, 2020.

---

> > ### Author Response · Authors · 2025-08-06
> > **Please let us know if any concerns remain — we’re happy to clarify**
> >
> > Dear Reviewer,
> >
> > I hope this message finds you well. As the discussion period is nearing its end with less than three days remaining, I want to ensure we have addressed all your concerns satisfactorily. If there are any additional points or feedback you'd like us to consider, please let us know. Your insights are invaluable to us, and we're eager to address any remaining issues to improve our work.
> >
> > Thank you for your time and effort in reviewing our paper.

---

### Decision · Program_Chairs · 2025-09-17

**Decision:**

Accept (poster)

**Comment:**

The paper proposes Dynamic Path Selection (DPS), a one-shot NAS strategy for semantic segmentation. DPS dynamically shifts training focus across convergence, expressiveness, and generalization using dedicated proxies (angle score, performance predictor, multi-scale information bottleneck). It integrates Pareto-based path selection, a dynamic stage scheduler, and diversity-driven exploration. Experiments on Cityscapes, CamVid, and BDD100K show DPS achieves state-of-the-art accuracy while maintaining efficiency, outperforming prior sampling- and gradient-based NAS methods.

Reviewers acknowledge the paper’s solid technical design, strong experimental results, and practical improvements in NAS for segmentation. Strengths include integration of multiple proxies, adaptive stage scheduling, and structural diversity maintenance. Weaknesses raised include limited theoretical novelty, computational overhead (especially from MS-IB), and sensitivity to hyperparameters. Some reviewers felt the contributions were more of a well-engineered integration than a fundamentally novel idea. Overall, three out of four reviewers leaned toward (borderline) acceptance due to empirical strength despite theoretical concerns.

Therefore, strong empirical results and well-motivated design outweigh concerns about novelty and overhead; the work might provide meaningful practical advances in NAS for semantic segmentation.
As the AC, I judge that this paper could be accepted.